# *Helicobacter pylori* Eradication According to Sequencing-Based 23S Ribosomal RNA Point Mutation Associated with Clarithromycin Resistance

**DOI:** 10.3390/jcm9010054

**Published:** 2019-12-25

**Authors:** Seung In Seo, Byoung Joo Do, Jin Gu Kang, Hyoung Su Kim, Myoung Kuk Jang, Hak Yang Kim, Woon Geon Shin

**Affiliations:** 1Department of Internal Medicine, Kangdong Sacred Heart Hospital, Hallym University College of Medicine, Seoul 05355, Korea; doctorssi@kdh.or.kr (S.I.S.); mmjjmj@naver.com (B.J.D.); bjpgr@kdh.or.kr (J.G.K.); hskim@kdh.or.kr (H.S.K.); mkjang@kdh.or.kr (M.K.J.); bacter@kdh.or.kr (H.Y.K.); 2Institute for Liver and Digestive Diseases, Hallym University, Chuncheon 24253, Korea

**Keywords:** *Helicobacter pylori*, clarithromycin, resistance, treatment

## Abstract

Background/Aims: Clarithromycin resistance in *Helicobacter pylori* is associated with point mutations in the 23S ribosomal RNA (rRNA) gene. We investigated the point mutations in the 23S rRNA genes of patients with clarithromycin-resistant *H. pylori* and compared the *H. pylori* eradication rates based on the point mutations. Methods: A total of 431 adult patients with *H. pylori* infection were recruited in Kangdong Sacred Heart Hospital in 2017 and 2018. Patients who did not have point mutations related to clarithromycin resistance and/or had clinically insignificant point mutations were treated with PAC (proton pump inhibitor, amoxicillin, clarithromycin) for seven days, while patients with clinically significant point mutations were treated with PAM (proton pump inhibitor, amoxicillin, metronidazole) for seven days. *H. pylori* eradication rates were compared. Results: Sequencing-based detection of point mutations identified four mutations that were considered clinically significant (A2142G, A2142C, A2143G, A2143C). The clarithromycin resistance rate was 21.3% in the overall group of patients. A2143G was the most clinically significant point mutation (84/431, 19.5%), while T2182C was the most clinically insignificant point mutation (283/431, 65.7%). The overall *H. pylori* eradication rate was 83.7%, and the seven-day PAM-treated clarithromycin-resistance group showed a significantly lower eradication rate than the seven-day PAC-treated nonresistance group (ITT; 55.4% (51/92) vs. 74.3% (252/339), *p* = 0.001, PP; 66.2% (51/77) vs. 88.4% (252/285), *p* = 0.0001). Conclusions: There were significantly lower eradication rates in the patients with clarithromycin-resistant *H. pylori* when treated with PAM for seven days. A future study comparing treatment regimens in clarithromycin-resistant *H. pylori*-infected patients may be necessary.

## 1. Introduction

*Helicobacter pylori* is a gram-negative bacterium that causes gastritis, peptic ulcer, gastric cancer, and mucosa-associated lymphoid tissue (MALT) lymphoma [1]. *H. pylori* gastritis is considered an infectious disease [1,2], and recent meta-analyses have proven that *H. pylori* eradication can significantly improve gastric atrophy [3,4]. Furthermore, increasing evidence has shown that *H. pylori* eradication might prevent the development of gastric cancer [5,6,7].

Until now, first line therapy for *H. pylori* eradication in Korea involves a regimen of a proton pump inhibitor (PPI), clarithromycin, and amoxicillin for 7–14 days [8]. However, there has been an increase in the rate of antibiotic resistance in the past two decades, leading to decreased *H. pylori* eradication rates by standard triple therapy [9]. Clarithromycin resistance associated with a 23S ribosomal RNA (rRNA) point mutation is the most crucial factor in the failure of *H. pylori* eradication. In Korea, the rate of resistance to clarithromycin was reported to be 5.9% before 2000 and increased to 21.4% in the period between 2006 and 2008 [10,11].

For this reason, the European guideline states that clarithromycin-containing triple therapy without susceptibility testing should be abandoned when the clarithromycin resistance rate in the region is higher than 15% [1]. Therefore, tailored *H. pylori* eradication by performing an antibiotic susceptibility test before treatment may be the most reasonable method to reduce the treatment failure; however, the process of bacterial culture is very lengthy and difficult, thus presenting limitations for clinical use.

Recently, there has been increasing evidence of molecular methods for the detection of *H. pylori* resistance to antibiotics such as clarithromycin and quinolones, which have been commonly used until now [12,13,14]. Dual priming oligonucleotide (DPO)-based multiplex polymerase chain reaction (PCR) assay detects mutational changes in *H. pylori* obtained from biopsy samples or feces. The method is relatively simple and accurate in finding minimal traces of genotypic resistant strains; however, it can only detect A2143G and A2142G mutations [12,13,14].

Clarithromycin-resistant strains frequently have several point mutations in the 23S rRNA gene, and A-to-G point mutations at positions 2142 and 2143 within the V domain have been demonstrated to be the main mutations; however, little is known regarding the clinical significance of the other point mutations [15,16]. DNA sequencing can detect more point mutations than DPO-PCR-based methods. There have been a few reports that investigated the clarithromycin resistance of *H. pylori* by sequencing-based detection of 23S rRNA gene point mutations.

The detailed point mutations of clarithromycin-resistant *H. pylori* have yet to be elucidated, and effective treatment strategies for areas of high clarithromycin resistance need to be developed. Consequently, this study aimed to investigate point mutations in patients infected with clarithromycin-resistant *H. pylori* and to compare the *H. pylori* eradication rates based on the clinically significant point mutations identified by 23S rRNA gene sequencing.

## 2. Methods

### 2.1. Patients

A total of 431 patients, ranging from 19 to 85 years old, were recruited in Kangdong Sacred Heart Hospital from January 2017 to September 2018. Subjects underwent upper endoscopy and were confirmed to have *H. pylori* infection by rapid urease test (CLOtest, Pronto Dry New; Medical Instruments Corp, Herford, Germany). The diagnosis of *H. pylori* was confirmed by a typical change in color during the rapid urease test. Mucosal biopsies taken from the antrum and body were used to detect clarithromycin resistance (CAMR)-related point mutations in patients who were positive in the rapid urease test.

Patients were excluded if they had received *H. pylori* eradication therapy within 1 year, antibiotics within 4 weeks, and surgery for gastric cancer, malignant tumors other than gastric cancer, severe systemic disorder such as end-stage renal disease or liver cirrhosis, or pregnancy. This study was approved by the ethics committee (2017-03-005, 3/28/2017) of Kangdong Sacred Heart Hospital, and the protocol conformed with the ethical guidelines of the Declaration of Helsinki (IRB no. 2017-03-005).

### 2.2. Detection of CAMR-Related Point Mutations 

*H. pylori* genomic DNA was isolated from a frozen gastric biopsy specimen that was stored at a temperature of less than −20 °C using the MagNA Pure 96 system (Roche Diagnostics Inc., Rotkreuz, Switzerland) and Viral NA SV Kit (Roche Diagnostics Inc., Indianapolis, IN, USA) according to the manufacturer’s instructions. PCR was conducted in a final reaction volume of 20 µL containing 4 µg of DNA, 2 µL of primer mixture, and 8 µL of 2× Master Mix (Samkwang Medical Laboratories, Seoul, South Korea). After an initial incubation step at 95 °C for 10 min, 45 amplification cycles were performed in AB SimpliAMP PCR (Applied Biosystems Inc., Foster City, CA, USA), using the following amplification parameters: 95 °C for 20 s, 55 °C for 30 s, and 72 °C for 30 s. The final extension was performed at 72 °C for 5 min.

Nucleotide sequencing of the amplified DNA was performed using ABI 3730 DNA analyzer (Applied Biosystems Inc., Foster City, CA, USA) with BigDye^®^ Terminator V3.1 (Applied Biosystems Inc., Foster City, CA, USA) according to the manufacturer’s instructions. All endpoint PCR reactions, agarose gel electrophoresis, and sequencing work were performed by Samkwang Medical Laboratories (SML, Seoul, Korea). This method can identify mutations in the nucleotide sequence of domain V in the 23S rRNA gene of *H. pylori* by amplifying the first 300 bp of the gene in seven *H. pylori* strains using PCR primers 23S F (5′-CGT AAC TAT AAC GGT CCT AAG-3′, corresponding to *H pylori* 23S rRNA gene positions 2007–2027) and 23S R (5′-TTA GCT AAC AGA AAC ATC AAG-3′, positions 2281–2301) to detect the mutations at positions 2115, 2141, 2142, 2143, 2144, 2147, 2182, 2190, 2195, and 2223.

### 2.3. Analysis of CAMR-Related Point Mutations

We analyzed the distribution of 23S rRNA point mutations associated with clarithromycin resistance detected by sequencing. From a previous study, a PCR-based method mainly detected the point mutations A2143G and A2142G. In Korea, there has been little information about other point mutations; therefore, we regarded A-to-G or A-to-C point mutations at positions 2142 and 2143 as clinically significant point mutations, while other mutations were regarded as clinically insignificant point mutations.

### 2.4. Study Design and Treatment Regimen

The flow diagram of this study is shown in Figure 1. Eradication of *H. pylori* was tailored to the detected CAMR-related point mutations. A2143C, A2143G, A2142G, and A2142C were regarded as clinically significant point mutations. The clarithromycin-resistant group was treated with PPI twice a day, amoxicillin (1000 mg) twice a day, and metronidazole (500 mg) three times a day for 7 days. Other mutations were regarded as clinically insignificant and were thus treated with the same regimen as the nonresistance group, which was treated with PPI twice a day, amoxicillin (1000 mg) twice a day, and clarithromycin (500 mg) twice a day for 7 days.

### 2.5. Outcome Assessment

The ^13^C-urease breath test (^13^C-UBT; UBiT-IR 300, Otsuka Pharmaceutical Co., Ltd., Tokyo, Japan), at least 6–8 weeks after completion of treatment, was performed to confirm *H. pylori* eradication. We defined successful eradication of *H. pylori* as a negative ^13^C-UBT result. All subjects were asked to stop taking PPI at least 2 weeks before ^13^C-UBT. Compliance was considered low when <90% of the prescribed pills were taken. Patients who did not visit for ^13^C-UBT after completion of treatment and presented low compliance were excluded from per-protocol (PP) analysis. 

### 2.6. Statistical Analysis 

Continuous variables were calculated as numbers and percentages. *H. pylori* eradication rates were analyzed based on intention-to-treat (ITT) and PP analyses. The eradication rates were compared by the chi-square test. A 2-sided *p* value less than 0.05 was considered statistically significant. All statistical analyses were carried out using SPSS for Windows, Version 19.0 (IBM Corp., Armonk, NY, USA). 

## 3. Results

### 3.1. Clinical Characteristics of Study Patients 

We recruited 431 consecutive patients who were positive for the rapid urease test and examined them for CAMR-related point mutations. Baseline characteristics of patients with tailored therapy (*n* = 362) are described in Table 1. Mean age was 54.7 ± 13.6 (19–85) years, and 65.2% of the subjects were male. Among the patients, 272 patients had peptic ulcer disease. Moreover, 48 patients had gastric adenoma, and 18 patients had gastric adenocarcinoma (Table 1).

### 3.2. Distribution of CAMR-Related Point Mutations

Among the 431 patients enrolled into the study, fifty isolates of *H. pylori* were not detected with any mutations. However, 21.3% of the enrolled patients had strains of *H. pylori* with clinically significant CAMR-related point mutations. A2143G was the most common point mutation, and there were patients who had both A2143G and A2142G mutations. Detailed information regarding the mutations is shown in Table 2. Other detected mutations were regarded as clinically insignificant. Among them, T2182C was the most common, and the mutations A2223G, T2190C, and C2195T were detected in six patients (Table 2).

### 3.3. Efficacy of Seven-Day H. pylori Eradication Regimen Tailored to Detected CAMR-Related Point Mutations

We assessed the efficacy of a seven-day *H. pylori* eradication regimen tailored to detected CAMR-related point mutations in 362 patients. Based on ITT analysis, the seven-day proton pump inhibitor, amoxicillin, metronidazole (PAM)-treated clarithromycin-resistance group showed a significantly lower eradication rate than the seven-day PAC-treated nonresistance group (55.4% (51/92) vs. 74.3% (252/339), *p* = 0.001) (Figure 2). Based on PP analysis, there was a significant difference between the two groups (66.2% (51/77) vs. 88.4% (252/285), *p* = 0.0001) (Figure 2). No serious adverse events were reported in both groups.

### 3.4. H. Pylori Eradication Rates According to CAMR-Related Point Mutations

We analyzed *H. pylori* eradication rates according to CAMR-related point mutations. The overall *H. pylori* eradication rate was 83.7% (Table 3). Among the clinically significant point mutations, A2142G showed the lowest eradication rate (1/4, 25%). Additionally, the eradication rate of the PAC treatment group, in which the patients presented no mutations, was 83.3% (Table 3).

## 4. Discussion

To our knowledge, this was the first study to compare the efficacy of *H. pylori* eradication regimens based on the 23S rRNA point mutations related to clarithromycin resistance identified through sequencing. Maastricht V recommendations state that susceptibility testing should be performed prior to therapy in regions with high clarithromycin resistance rates [17]. A recent meta-analysis demonstrated that susceptibility-guided therapy was superior to empirical therapy [18]. Antimicrobial susceptibility-guided therapy as first-line therapy may be more logical to avoid misuse of antibiotics in the *H. pylori* treatment, considering increasing antibiotic resistance. However, culture-based methods are time-consuming (10–14 days) and require specific expertise, making their application difficult in clinical practice as first-line therapy. There have been several studies about tailored *H. pylori* treatment based on molecular methods [19,20,21]. In most studies, only mutations A2143G and A2142G were detected by DPO-based multiplex PCR. Our study aimed to detect more point mutations other than A2143G and A2142G, as well as investigate the clinical significance of these mutations in treating *H. pylori*.

Among the point mutations identified, 21.3% were clinically significant. Among those, A2143G was the most common point mutation (19.5%). The same result was observed in previous studies [19,20,21]. Although one patient had the clinically significant A2142C mutation, T2182C was the most common clinically insignificant mutation as analyzed via sequencing. There have been controversial results regarding whether T2182C is associated with clarithromycin resistance [10,15]. Of note, the T2182C mutation is the most frequent point mutation associated with clarithromycin resistance in Korea; however, its clinical significance has not been determined [15,22,23]. A sequencing-based method can detect more point mutations, including T2182C, than other methods. The factors influencing treatment failure for clinically insignificant CAMR-related point mutations should be clarified in the future.

In the present study, there was a significantly lower eradication rate in the seven-day PAM treatment group of clarithromycin resistance compared with that in the seven-day PAC treatment group of negative clarithromycin resistance (ITT; 55.4% (51/92) vs. 74.3% (252/339), *p* = 0.001, PP; 66.2% (51/77) vs. 88.4% (252/285), *p* = 0.0001). The lower eradication rate in the mutant type may be explained by the shorter treatment duration and dual resistance to clarithromycin and metronidazole. Similarly, a recent Korean study regarding tailored *H. pylori* eradication regimens based on a DPO-based PCR method found that there were significant differences in the *H. pylori* eradication rates depending on the presence of a 23S rRNA gene point mutation (mutant type vs. wild type; 81.8% (45/55) vs. 94.9% (131/138), *p* = 0.004). In this study, a seven-day PAM regimen was used to treat clarithromycin-resistant *H. pylori*, similarly to our study. Furthermore, the lower eradication failure might be due to amoxicillin resistance [24]. Although we could not investigate amoxicillin resistance, recent studies, including a Korean nationwide study, reported the resistance rate of amoxicillin as 9.5% [25].

It has not been fully agreed which regimen should be used in patients for primary treatment of clarithromycin-resistant *H. pylori*. In recent tailored therapy according to the DPO-based PCR method, 14-day bismuth-based quadruple therapy in the clarithromycin-resistant group showed a higher eradication rate (33/36, 91.7%) [19]. Although bismuth-based quadruple therapy showed a higher eradication rate for clarithromycin-resistant *H. pylori*, considering the side effects of administering multiple drugs and the increasing incidence of antibiotic resistance, metronidazole-based triple therapy may be an appropriate alternative therapy. Resistance to metronidazole has increased worldwide; however, in Korea, a decreasing trend in metronidazole resistance was observed in several studies [10]. In Korea, the expected rate of dual resistance is lower than 15%. Furthermore, metronidazole resistance is known to be partially overcome by increasing the dose and treatment duration [1]. Thus, 14-day PAM treatment may be applied for clarithromycin-resistant *H. pylori* treatment.

In this study, we regarded A-to-G or A-to-C point mutations at positions 2142 and 2143 as clinically significant, while other mutations were regarded as clinically insignificant. Previous studies using the PCR-Restriction Fragment Length Polymorphism (RFLP) system have shown the high frequency of the A2143G or A2144G mutations in clarithromycin-resistant *H. pylori* in both the United States and Europe [16,26]. There was no mutation at position 2144 in our study, indicating that the main 23S rRNA gene mutations inducing clarithromycin resistance are dissimilar in western and eastern countries [14]. Moreover, we documented that clinically insignificant point mutations had little influence on the *H. pylori* eradication rate; however, this has yet to be confirmed and fully elucidated.

Considering the increasing incidence of antibiotic resistance, tailored *H. pylori* eradication regimens based on clarithromycin resistance detected via molecular methods for primary treatment may be more cost-effective than empirical treatment [21]. There are few Korean studies that have analyzed the detailed CAMR-related point mutations identified by a sequencing-based method in the era of high clarithromycin resistance. Furthermore, the efficacy of seven-day PAM treatment in primary clarithromycin-resistant *H. pylori* was not determined. To our knowledge, our study is the first relatively large-scale research regarding the tailored therapy of *H. pylori*, based on CAMR-related point mutations identified by sequencing of the 23S rRNA gene.

This study has some limitations. First, we could not perform *H. pylori* culture and antibiotic susceptibility testing [14]. It is reported that there may be discordance between genotype and phenotype [27]. A future study comparing molecular-based and culture-based methods may be necessary. Second, this was a retrospective study; thus, we could not analyze other factors influencing *H. pylori* eradication rates other than clarithromycin resistance. Notably, among the 42 patients who had no point mutations and were treated with a seven-day PAC regimen, only 35 patients were fully cleared of *H. pylori* infection. This means that there may be complex factors associated with eradication failure, such as cytochrome P450 2C19 (CYP2C19) polymorphism influencing PPI metabolism. Lastly, we could not compare tailored therapy with empirical PAC treatment, as there was a small portion of patients who were treated empirically without a CAMR mutation test during the study period. However, several recent studies have already shown the better efficacy of tailored therapy compared with empirical treatment, and therefore, there could be ethical issues. 

In conclusion, our study identified multiple point mutations in the 23S rRNA gene as identified by sequencing and that a seven-day metronidazole-based regimen may be less efficacious in treating clarithromycin-resistant *H. pylori* infection. Therefore, a prospective study comparing different treatment regimens may be needed in optimizing clarithromycin-resistant *H. pylori* treatment. 

## Figures and Tables

**Figure 1 jcm-09-00054-f001:**
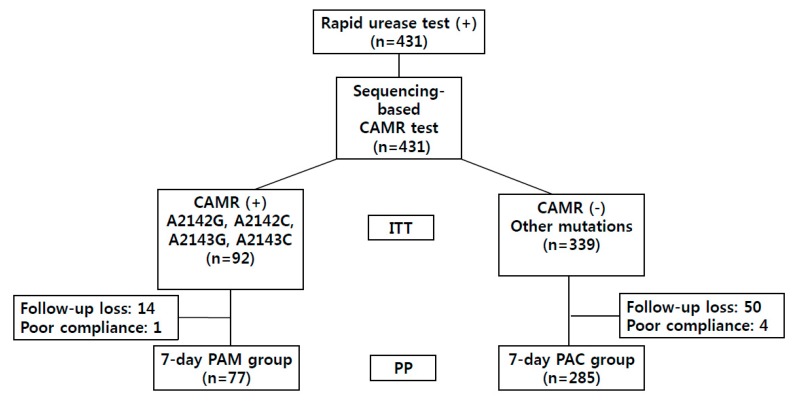
Flow chart of the study. CAMR, clarithromycin resistance; PAM, proton pump inhibitor, amoxicillin, and metronidazole; PAC, proton pump inhibitor, amoxicillin, and clarithromycin; PP, per-protocol analysis; ITT, intention-to-treat analysis.

**Figure 2 jcm-09-00054-f002:**
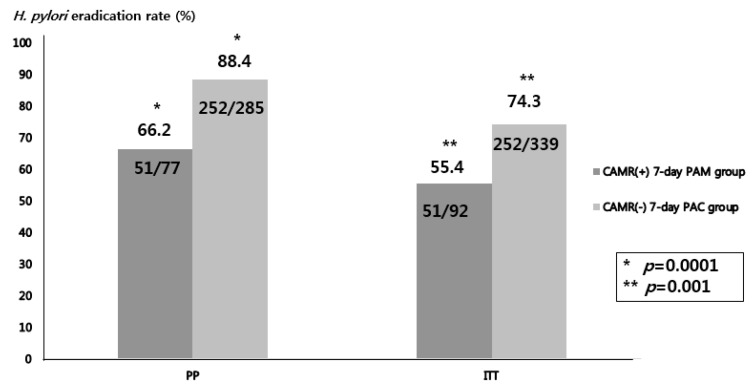
The *Helicobacter pylori* eradication rate according to clarithromycin resistance. CAMR, clarithromycin resistance; PAM, proton pump inhibitor, amoxicillin, and metronidazole; PAC, proton pump inhibitor, amoxicillin, and clarithromycin; PP, per-protocol analysis; ITT, intention-to-treat analysis.

**Table 1 jcm-09-00054-t001:** Baseline characteristics of patients with tailored therapy (*n* = 362).

Variables	
Age, mean ± SD (range)	54.7 ± 13.6 (19–85)
Male, *n* (%)	236 (65.2)
Clinical diagnosis, n (%)	
Nodular gastritis	16 (4.4)
Chronic atrophic gastritis	8 (2.2)
Peptic ulcer	272 (75.1)
Gastric adenoma	48 (13.3)
Gastric adenocarcinoma	18 (5)

**Table 2 jcm-09-00054-t002:** Distribution of clarithromycin resistance-related point mutations determined through sequencing (*n* = 431).

Point Mutation	Number (%)
Clinically significant mutation	92 (21.3)
A2142G	4 (0.9)
A2142C	1 (0.2)
A2143G	84 (19.5)
A2142G + A2143G	3 (0.7)
Clinically insignificant mutation	289 (67.1)
T2182C	283 (65.7)
others	6 (1.4)
No point mutation	50 (11.6)

**Table 3 jcm-09-00054-t003:** *H. pylori* eradication rates according to clarithromycin resistance-related point mutations (*n* = 362).

Point Mutation	Eradication Rate (%)
Overall	303/362 (83.7)
none	35/42 (83.3)
A2142G	1/4 (25)
A2142C	1/1 (100)
A2143G	47/70 (67.1)
A2142G+A2143G	2/2 (100)
T2182C	212/238 (89.1)
Others (A2223G, C2195T)	5/5 (100)

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
