# Peer review of "Helicobacter pylori Eradication According to Sequencing-Based 23S Ribosomal RNA Point Mutation Associated with Clarithromycin Resistance"

_jcm, 2019, doi:10.3390/jcm9010054_

Round 1
Reviewer 1 Report
The article „Helicobacter pylori eradication according to sequencing-based 23S ribosomal RNA point mutation associated with clarithromycin resistance” attempts to show the relationship between the occurrence of genetic mutations and H. pylori resistance to clarithromycin. In addition, authors decided to check the effectiveness of two therapies (containing metronidazole or clarithromycin) in the eradication of infections caused by isolates having the mutations studied. Because WHO listed clarithromycin-resistant H. pylori isolates as one of the most important pathogens for which new methods of eradication should be sought, I consider the current topic important.
The article is written correctly, in a comprehensible way, with the occurrence of some small mistakes. Examples of suggested corrections:
“the 7-day PAM-treated clarithromycinresistance group showed” -> the 7-day PAM-treated clarithromycin-resistance group showed [Abstract] helicobacter pylori -> Helicobacter pylori [Key words] “A total of 431 patients were recruited in Kangdong Sacred Heart Hospital from January 2017 to September 2018. Subjects who underwent upper endoscopy and were confirmed of H. pylori infection by rapid urease test (CLOtest, Pronto Dry New; Medical Instruments Corp, Herford, Germany) ranged from 19–85 years old.” -> A total of 431 patients, ranged from 19–85 years old, were recruited in Kangdong Sacred Heart Hospital from January 2017 to September 2018. Subjects who underwent upper endoscopy and were confirmed of H. pylori infection by rapid urease test (CLOtest, Pronto Dry New; Medical Instruments Corp, Herford, Germany). [Methods] “13C-urease breath test (13C-UBT; UBiT-IR 300, Otsuka Pharmaceutical Co., Ltd, Tokyo, Japan) at least 6–8 weeks ...” -> The 13C-urease breath test (13C-UBT; UBiT-IR 300, Otsuka Pharmaceutical Co., Ltd, Tokyo, Japan) at least 6–8 weeks … [Methods] “50 patients were not detected of any mutations.” -> Fifty isolates of H. pylori were not detected with any mutations. [Results] “However, 21.3% of the enrolled patients had clinically significant CAMR-related point mutations.” -> However, 21.3% of the enrolled patients had strains of H. pylori with clinically significant CAMR-related point mutations. [Results] Also in other places in the text: bacteria have these mentioned mutations and not patients (please change all) “to detected CAMRrelated point mutations” -> to detected CAMR-related point mutations [Results] “Helicobacter pylori eradication rate” -> The Helicobacter pylori eradication rate [Figure 2] “Maastricht recommendations state that susceptibility testing should be” -> Maastricht V recommendations state that susceptibility testing should be [Discussion] “avoid misuse of antibiotics in H. pylori treatment in the era of…” -> avoid misuse of antibiotics in the H. pylori treatment in the era of [Discussion] “However, culturebased methods are time-consuming” -> However, culture-based methods are time-consuming [Discussion] “the increasing incidence of antibiotic resistance, metronidazolebased triple therapy may be an appropriate alternative therapy” -> the increasing incidence of antibiotic resistance, metronidazole-based triple therapy may be an appropriate alternative therapy [Discussion] References: 2,3,4,8,9,10,11,15,16,18,19,20,23, and 25 -> a full range of pages is needed, e.g. 283-286 (not 283-6)References: 8,12,16, and 21 -> abbreviations of journals are needed
Author Response
Thank you for your kind revision. We corrected the manuscript as your recommendation. I highlighted the revised contents as yellow color.
We appreciate about your kind contribution again.
Reviewer 2 Report
Major revision
Although there were significantly lower eradication rates in the patients with clarithromycin-resistant H. pylori when treated with PAM for 7 days, the determinant of eradication failure is not only due to clarithromycin resistance, but also due to amoxicillin resistance, even though its effect is very small. The authors need to address this issue such as antibiotic resistance other than clarithromycin. In that case, the authors need to mention about the previous data for amoxicillin resistance or non-sensitive. The authors need to revise extensively including the above information.
Not only for CAM resistance, but also AMPC resistance should be considered for the assessment. (Nishizawa T, Suzuki H, Tsugawa H, Muraoka H, Matsuzaki J, Hirata K, Ikeda F, Takahashi M, Hibi T. Enhancement of amoxicillin resistance after unsuccessful Helicobacter pylori eradication. Antimicrob Agents Chemother. 2011 Jun;55(6):3012-4. doi: 10.1128/AAC.00188-11. Epub 2011 Apr 12. Erratum in: Antimicrob Agents Chemother. 2013 Feb;57(2):1106. PubMed PMID: 21486961; PubMed Central PMCID: PMC3101459.)
These notions should be included in the measurement.
Author Response
Thank you for your kind review.
We included your point to DISCUSSION as follows with the reference you recommended and recent Korean paper.
Also, the lower eradication failure might be due to amoxicillin resistance24. Although, we could not investigate amoxicillin resistance, recent studies including Korean nationwide study reported resistance rate of amoxicillin as 9.5%25.
Also, we received English editing by Editage, and I attach the certificate from Editage.
We checked our manuscript and corrected throughout the paper.
Thank you for your help again.

Round 2
Reviewer 2 Report
It has been well revised.